# Gα_i2_ Signaling Regulates Neonatal Respiratory Adaptation

**DOI:** 10.3390/ijms262110655

**Published:** 2025-11-01

**Authors:** Veronika Leiss, Katja Pexa, Andreas Nowacki, James P. Bridges, Benedikt Duckworth-Mothes, Susanne Ammon-Treiber, Ana Novakovic, Franziska Zeyer, Hartwig Wolburg, Petra Fallier-Becker, Roland P. Piekorz, Matthias Schwab, Letizia Quintanilla-Martínez, Sandra Beer-Hammer, Bernd Nürnberg

**Affiliations:** 1Department of Pharmacology, Experimental Therapy and Toxicology, Institute of Experimental and Clinical Pharmacology and Pharmacogenomic, Interfaculty Center for Pharmacogenomics and Drug Research, Eberhard Karls University Tübingen and University Clinic, 72076 Tübingen, Germany; veronika.leiss@uni-tuebingen.de (V.L.); andreas.nowacki@insel.ch (A.N.); benedikt.mothes@med.uni-tuebingen.de (B.D.-M.); anovakovic@ukaachen.de (A.N.); 2Institute for Biochemistry and Molecular Biology II, Medical Faculty, Heinrich Heine University Düsseldorf, 40225 Düsseldorf, Germany; katja.pexa@zuv.uni-hannover.de (K.P.); roland.piekorz@uni-duesseldorf.de (R.P.P.); 3Department of Medicine, National Jewish Health, Denver, CO 80222, USA; bridgesj@njhealth.org; 4Department of Medicine, Division of Pulmonary Sciences and Critical Care Medicine, University of Colorado Anschutz Medical Campus, Aurora, CO 80045, USA; 5Institute of Clinical Anatomy and Cell Analysis, University of Tübingen, 72076 Tübingen, Germany; 6Department of Pharmacology, Toxicology and Clinical Pharmacy, Institute for Pharmacy, University of Tübingen, 72076 Tübingen, Germany; susanne.ammon-treiber@uni-tuebingen.de (S.A.-T.);; 7Interdisciplinary Paediatric Intensive Care Unit 34, Clinic for Paediatrics and Adolescent Medicine, University Clinic Tübingen, 72016 Tübingen, Germany; franziska.zeyer@med.uni-tuebingen.de; 8Institute of Pathology and Neuropathology, Comprehensive Cancer Center, Eberhard Karls University Tübingen and University Clinic, 72074 Tübingen, Germany; hartwig.wolburg@uni-tuebingen.de (H.W.); petra.fallier-becker@uni-tuebingen.de (P.F.-B.); leticia.quintanilla-fend@med.uni-tuebingen.de (L.Q.-M.); 9Dr. Margarete Fischer-Bosch Institute of Clinical Pharmacology, 70376 Stuttgart, Germany; 10Department of Clinical Pharmacology, Institute of Experimental and Clinical Pharmacology and Pharmacogenomic, Interfaculty Center for Pharmacogenomics and Drug Research, Eberhard Karls University Tübingen and University Clinic, 72074 Tübingen, Germany

**Keywords:** Gα_i_-signaling, Gα_i2_, lung development, surfactant, respiratory distress syndrome (RDS), neonatal lethality

## Abstract

Heterotrimeric G_i_ proteins are crucial modulators of G protein-coupled receptor signaling, with Gα_i2_ ubiquitously expressed and implicated in diverse physiological processes. Previous reports described partial lethality in *Gnai2*-deficient mice, but the timing and mechanism of death remained unclear. Here, we demonstrate that impaired neonatal respiratory adaptation contributes to mortality in *Gnai2*-deficient neonates. Despite normal Mendelian distribution at birth and no overt malformations, at least 20% of the expected *Gnai2*-deficient neonates died within minutes after birth, displaying abnormal breathing, cyanosis, and features resembling neonatal respiratory distress syndrome (RDS). Histological and ultrastructural analyses revealed reduced alveolar surface area, thickened septa, increased mesenchymal tissue, and impaired surfactant ultrastructure, despite unaltered alveolar surfactant phospholipid levels. These findings suggest that Gα_i2_ modulates the structural deployment and functional organization of surfactant within alveoli, although the incomplete phenotype and survival of some neonates indicate a regulatory rather than indispensable role of Gα_i2_. Our data underscore the complexity of neonatal respiratory adaptation and highlight potential systemic and intercellular mechanisms underlying alveolar stabilization.

## 1. Introduction

Heterotrimeric G proteins (Gαβγ) are pivotal signal transducers activated by G protein-coupled receptors (GPCRs) and other regulatory proteins [for review see [1,2,3,4]]. They mediate the selective coupling of an impressive variety of membrane receptors to intracellular effector systems, thereby eliciting specific cellular responses. This functional specificity is conferred by the diversity of G protein isoforms [5,6]. The Gα subunits are subdivided into families based on sequence homology and functional properties and determine receptor coupling specificity [7]. In contrast, while Gα subunits directly regulate only a limited number of cellular effectors, Gβγ dimers are primarily responsible for selecting and modulating downstream effectors such as enzymes, ion channels, and transporters [8].

G_i_ proteins, whose Gα subunits exist in three isoforms (Gα_i1_, Gα_i2_, Gα_i3_), are widely expressed and transmit signals from numerous GPCRs involved in development and maintaining homeostasis of the organism [9]. Studies using genetically modified mouse models lacking individual Gα_i_ proteins have provided important insights into their physiological and pathophysiological roles [10,11,12,13,14]. G_i_ proteins contribute to critical functions in the cardiovascular, endocrine and immune systems, the central nervous system, and in sensory organs [15,16,17,18,19,20,21,22,23,24,25,26].

Among the three G_i_ isoforms, Gα_i2_ is unique in being ubiquitously expressed and highly abundant across tissues. Although global deletion of the *Gnai2* gene does not prevent embryonic development, the initial characterization of *Gnai2*-deficient mice indicated a significantly reduced number of homozygous offspring by postnatal day 28 [13]. A follow-up study confirmed that genotype distribution at embryonic day 14 was as expected, but only approximately half of *Gnai2*-deficient pups survived to postnatal day 21, suggesting increased lethality [10]. This phenotype contrasts with that of mice deficient in Gα_i1_ or Gα_i3_, which do not exhibit comparable mortality [10,12]. The underlying cause and timing of death in *Gnai2*-deficient mice remain unclear. Interestingly, mice expressing a constitutively active knock-in mutant of Gα_i2_ also show increased perinatal mortality, indicating that both loss and dysregulation of Gα_i2_ activity can compromise neonatal survival [27].

Potential causes of lethality during the prenatal, perinatal, and early postnatal periods in *Gnai2*-deficient mice may stem from maternal factors—such as impaired uterine contractility or placental insufficiency—or from fetal abnormalities. The latter may involve cardiovascular or neurological defects, immune dysfunction, or impaired lung development resulting in insufficient respiratory adaptation [14,28].

The aim of this study was to elucidate the role of Gα_i2_ in the perinatal mortality observed in a subset of *Gnai2*-deficient mice. To this end, we addressed two critical questions: (1) at what developmental stage do *Gnai2*-deficient mice die, and (2) what the underlying cause of death is. Our findings demonstrate that a significant proportion of *Gnai2*-deficient mice die within minutes after birth. Morphological analysis of lung tissue revealed reduced alveolar expansion and altered alveolar surfactant ultrastructure, consistent with impaired respiratory function and early postnatal respiratory failure of *Gnai2*-deficient mice. Interestingly, isolated lungs from *Gnai2*-deficient mice reaching adulthood revealed normal or even slightly improved lung function, which might be caused by a compensatory upregulation of the quantitatively minor expressed Gα_i3_ isoform.

## 2. Results

### 2.1. Gnai2-Deficient Mice Exhibit Early Perinatal Lethality

In a previous study, we reported that the survival rate of *Gnai2*-deficient mice was reduced by approximately 50% between embryonic day 14 (E14) and postnatal day 21 (P21) [10]. To further pinpoint the window of increased mortality, we analyzed offspring from heterozygous intercrosses (*Gnai2*^+/−^ × *Gnai2*^+/−^ breeding) at defined developmental stages: E14, E18, postnatal days 5–8 (P5–8), and P21. At both E14 and E18, the proportion of homozygous *Gnai2* knockout (*Gnai2*^−/−^) fetuses was consistent with Mendelian expectations of 25% (Table 1). However, by P5–8 and P21, *Gnai2*^−/−^ mice were markedly underrepresented, accounting for only 11% and 9.7% of the population, respectively. This was accompanied by a reduction in average litter size compared to wild-type (wt) mating (Appendix A). No gender-specific differences were observed, nor was it dependent on the genetic background of the mice (i.e., C57BL/6 or Sv129) (Appendix A). Concurrently, *Gnai3*-deficient mice—previously shown not to exhibit increased lethality [10]—displayed normal Mendelian ratios at P21 and no reduced average litter sizes following heterozygous intercrosses (Appendix A).

Since birth poses a series of physiological challenges and risks, we speculated that the increased mortality observed in *Gnai2*-deficient mice might be caused perinatally. To address this, we first examined whether the expected number of *Gnai2*-deficient offspring was already reduced at the time of birth and whether *Gnai2*-deficient neonates were initially viable. To this end, we analyzed the number and vitality of neonates derived from *Gnai2*^+/−^ × *Gnai2*^–/–^ mating delivered by cesarean section at E19.5. A total of 78 neonates were obtained, of which 41 were heterozygous and 37 were homozygous *Gnai2*-deficient. This distribution was consistent with the expected Mendelian ratio (1:1), indicating that the frequency of *Gnai2*-deficient neonates at the time of birth was normal and that prenatal embryonic lethality was negligible. Moreover, all 78 neonates displayed clear signs of initial breathing behavior immediately after delivery. Heterozygous mice served as controls, as previous studies had shown no increased mortality in this genotype (see Table 1). As part of this initial exploratory study, we assessed the postnatal viability of *Gnai2*-deficient mice over a 2 h observation period. A total of 40.5% of homozygous *Gnai2*-deficient neonates died shortly after cesarean delivery. Unexpectedly, 14.6% of heterozygous control neonates also exhibited early postnatal lethality (Figure 1a,b), possibly reflecting limited procedural experience at this early stage of the project and/or increased vulnerability of pups delivered by cesarean section. Despite this, neonatal mortality in the knockout group was 25.9% higher than in heterozygous controls. Moreover, the average time to death was significantly shorter in *Gnai2*-deficient neonates (12.3 min) compared to heterozygous controls (17.8 min) (Figure 1c). Mean birth weight was similar across groups (1.0 ± 0.1 g), and no gross morphological abnormalities were observed.

Having identified the immediate postnatal period as critical for survival, we next aimed to clarify the underlying cause by analyzing spontaneously delivered *Gnai2*-deficient newborns and their wild-type (wt) littermates. Both groups were derived from the same breeding colony, with *Gnai2*-deficient neonates obtained from homozygous *Gnai2*^–/–^ × *Gnai2*^–/–^ mating and wt controls from homozygous *Gnai2*^+/+^ × *Gnai2*^+/+^ mating. This breeding strategy allowed us to obtain sufficient numbers of *Gnai2*-deficient offspring in a reasonable time frame.

Consistent with our initial observations, a subset of *Gnai2*-deficient neonates died shortly after birth, with 10 of 49 pups (20.4%) exhibiting early postnatal lethality during the 2 h observation period (Figure 1d, e). This lethality rate closely matched the 25.9% increase observed in our initial cohort compared with controls. Notably, all deceased *Gnai2*-deficient neonates displayed gasping at birth but failed to establish a regular breathing rhythm before death, in contrast to the survivors. The median survival time of *Gnai2*-deficient neonates that died postnatally was 22 min, with a mean survival time of 29 ± 21 min (Figure 1f). Among 17 wild-type controls, one stillbirth occurred, while all remaining neonates survived the full 2 h observation period.

Taken together, these data demonstrate that the frequency of *Gnai2*-deficient mice at birth is normal, but a significant proportion dies shortly thereafter, indicating early postnatal lethality. This is in line with reduced litter sizes at P21 from homozygous *Gnai2*-deficient mating compared not only to wt mating but also to heterozygous mating (see Appendix A).

The cause of death could not be attributed to any obvious macroscopic abnormalities. Somatometric and organometric analysis of the neonatal mice revealed no differences in crown–rump length between *Gnai2*-deficient and wt controls; however, body weights were significantly lower in *Gnai2*-deficient neonates (Figure 2a,b). Organ weights for the liver were comparable between groups, whereas organ weights for the kidney, lung, and heart were significantly lighter in *Gnai2*-deficient neonates (Figure 2c–f). To further explore potential causes of death, we analyzed body and organ weights separately in deceased (*Gnai2*^–/–^_(d)_) and surviving *Gnai2*-deficient (*Gnai2*^–/–^_(s)_) littermates. No differences in crown–rump length, body or specific organ weights were observed between deceased *Gnai2*^−/−^_(d)_ and surviving *Gnai2*^–/–^_(s)_ neonates (Appendix A).

### 2.2. Gnai2-Deficiency Results in Lung Dysfunction in a Subset of Mice

Under physiological conditions, the respiratory pattern of newborns undergoes a rapid transition within the first hour of life—from initial irregular, wheezing-like efforts to a stable and vigorous rhythm characterized by an increased respiratory rate. During this adaptation phase, the early wheezing persists transiently alongside deep inspiratory efforts that gradually become faster and more synchronized [29]. In contrast, all deceased *Gnai2*^–/–^ newborns exhibited the characteristic initial wheezing, but failed to establish a regular respiratory rhythm. Instead, these newborns continued to take deep breaths accompanied by subcostal and abdominal retractions. These were interrupted by increasingly long episodes of apnea (lasting up to one minute) and progressive cyanosis. This clinical presentation resembled neonatal respiratory distress syndrome (RDS) in humans [30,31]. However, close inspection of the affected *Gnai2*-deficient neonates revealed that lung borders and expansion were visible through the thin skin, suggesting that the lungs of those that later died were initially inflated (Figure 3a–c). In contrast, none of the wt neonates, nor the majority of surviving *Gnai2*-deficient neonates displayed overt abnormalities in breathing patterns. In these animals, the typical postnatal transition from initial deep gasps to a high-frequency breathing pattern occurred within minutes after delivery.

### 2.3. Gnai2-Deficiency Reduces Alveolar Surface Area in Deceased Newborn Mice

Our findings indicate that the elevated lethality of *Gnai2*-deficient offspring can be largely attributed to neonatal death occurring immediately after birth. This mortality appears to result from insufficient lung function consistent with features of neonatal respiratory distress syndrome (RDS). Potential causes of lung dysfunction include morphological abnormalities such as lung hypoplasia or congenital alveolar dysplasia, as well as surfactant deficiency or defects [32].

To address this, we examined lung development histologically in *Gnai2*-deficient embryos at embryonic days (E) 15.5 and 17.5, compared with wt controls. At E15.5, the lungs are in the glandular stage, where the bronchial tree forms and develops through extensive branching [33]. 

*Gnai2*-deficient embryos exhibited a cell-rich mesenchyme with numerous primitive bronchi and branching points (Figure 4a). The area fraction of primitive bronchi was significantly higher in *Gnai2*-deficient embryos (6.7 ± 0.6%, *n* = 7) than in wt controls (5.5 ± 0.5%, *n* = 7; *p* < 0.001; Figure 4b). By E17.5, this fraction increased to 14 ± 2.9% in *Gnai2*-deficient embryos (*n* = 7), now comparable to wt controls (13.5 ± 5.5%, *n* = 5; Figure 4d,e). At this stage, the lungs corresponded histologically to the canalicular phase, where the air–blood barrier begins to form and primitive epithelial sacs appear. The proportion of mesenchymal cell areas did not differ significantly between genotypes at either time point (Figure 4c,f).

Although embryonic lung development did not reveal persistent structural abnormalities in deceased *Gnai2*-deficient mice (Figure 5), hematoxylin and eosin (H&E) staining of paraffin-embedded lung tissue from newborns demonstrated marked differences in alveolar and mesenchymal areas (Figure 6a).

While alveolar areas were similar between wt and surviving *Gnai2*^−/−^_(s)_ neonates, deceased *Gnai2*^−/−^_(d)_ neonates exhibited significantly reduced alveolar areas compared with wt (Figure 6b). Furthermore, lungs of *Gnai2*^−/−^_(d)_ neonates appeared underdeveloped, with thickened interalveolar septa (Figure 6a, right). Consistently, the mesenchymal area was significantly increased in *Gnai2*^−/−^_(d)_ neonates (Figure 6a, right) compared to both wt (Figure 6a, left) and *Gnai2*^−/−^_(s)_ (Figure 6a, middle) neonates (Figure 6c).

### 2.4. Secreted Alveolar Surfactant Fails to Properly Unfold in Deceased Newborn Gnai2-Deficiency Mice

The observed breathing defect together with the histological findings in deceased *Gnai2*^−/−^ neonates prompted us to test the hypothesis if impaired surfactant secretion is the underlying cause of the respiratory failure. Notably, proper lung function at birth depends on adequate production and secretion of pulmonary surfactant. Surfactant is synthesized by alveolar type II (ATII) cells, stored in intracellular lamellar bodies (LBs), and secreted into the alveolar space, where it is essential for normal breathing immediately after birth [34,35]. A deficiency or functional impairment of surfactant results in alveolar collapse after initial inflation due to insufficient reduction in surface tension, along with increased mesenchyme-rich, compact lung tissue and thickened interalveolar septa—consistent with the observations in *Gnai2*^−/−^_(d)_ neonates (Figure 6). We therefore examined lungs from these animals by electron microscopy to assess LBs and the alveolar surfactant lattice structures (Figure 7). In WT mice, typical intracellular LBs and well-formed extracellular tubular myelin, representing structured surfactant reserves in the alveolar fluid, were readily identifiable (Figure 7a,b). A similar pattern was observed in *Gnai2*^−/−^_(s)_ neonates (Figure 7c,d). In contrast, *Gnai2*^−/−^_(d)_ neonates showed altered ultrastructure of alveolar surfactant unfolding and failure to form a phospholipid monolayer at the air–liquid interface, as evidenced by the absence or disruption of tubular myelin structures (Figure 7e,f).

The unfolding of LBs into a functional monolayer is facilitated by the hydrophobic surfactant proteins SP-B and SP-C, which promote lipid insertion and film stabilization [35]. In addition, SP-A, in a calcium-dependent manner, organizes and aggregates surfactant lipids into functional structures. To exclude a general surfactant deficiency as the cause, we quantified saturated phosphatidylcholine (SatPC), the most abundant surface-active phospholipid species in surfactant. In whole-lung tissue from *Gnai2*-deficient embryos at E18.5, SatPC levels did not differ from wt controls (Appendix A).

To determine whether loss of Gα_i2_ specifically in ATII or mesenchymal cells was sufficient to reproduce the phenotype, we generated transgenic mice with cell-specific deletion of Gα_i2_ using Cre recombinase driven by either the Shh promoter (ATII-specific) or the Dermo1 promoter (mesenchymal-specific). Shh-Cre mice and Dermo1-Cre mice have been used extensively in lung development research due to high deletion efficiency in pulmonary epithelial cells and lung mesenchymal cells, respectively [36,37,38,39,40]. Offspring from both lines exhibited normal survival after birth and showed no evidence of respiratory distress syndrome. Furthermore, double deficiency of Gα_i2_ and the closely related Gα_i3_ isoform in ATII cells unexpectedly resulted in the expected Mendelian ratios at birth and normal survival beyond 2 h postpartum (Appendix A). Consistent with these findings, preliminary analyses of SatPC and total phospholipid content in BALF and lamellar body homogenates from ATII-specific *Gnai2; Gnai3* double-deficient mice showed no significant differences compared to controls (Appendix A), excluding a primary defect in surfactant synthesis or secretion by these cells. Thus, potential extrapulmonary effects warrant further investigation to clarify the cause of the described postnatal lethality in global *Gnai2*-deficient mice.

### 2.5. Improved Lung Function in Adult Gnai2-Deficient Mice

To validate if the lack of Gα_i2_ in surviving *Gnai2*^−/−^ mice has any impact on lung function, we used the ex vivo isolated, perfused and ventilated lung model (IPL), which allows for measuring lung parameters under a constant pressure. Adult *Gnai2*^−/−^ animals displayed normal to improved lung function (Figure 8 and Appendix A).

While airway resistance and pulmonary artery pressure (PAP) were comparable between genotypes (Figure 8a,b), the inspiratory and expiratory flow rates, tidal volume, and compliance were significantly increased in *Gnai2*-deficient animals (Figure 8c–f).

Given the high homology and partially redundant functions of Gα_i_ isoforms within specific cell types, we hypothesized that upregulation of the closely related Gα_i3_ isoform might compensate for the loss of Gα_i2_. To examine this, lung homogenates were analyzed by immunoblotting for Gα_i2_ and Gα_i3_ expression. As expected, Gα_i2_ protein was readily detectable in WT lungs but absent in *Gnai2*-deficient tissue (Figure 8g,i). Interestingly, Gα_i3_ expression was significantly increased in lungs of *Gnai2*-deficient mice (Figure 8h,i). Although individual cell types were not analyzed in this study, these findings suggest a compensatory role of Gα_i3_ in maintaining proper lung function in adult *Gnai2*-deficient mice.

## 3. Discussion

Our findings demonstrate that the lack of Gα_i2_ compromises neonatal respiratory adaptation, resulting in increased early postnatal mortality. A substantial proportion of *Gnai2*-deficient neonates failed to establish regular breathing and died within minutes of birth, presenting with cyanosis and a clinical pattern reminiscent of neonatal respiratory distress syndrome (RDS) [30,31]. Importantly, these neonates were born at expected Mendelian ratios, exhibited normal vitality at delivery, and had no gross morphological abnormalities, indicating that the lethality is based on postnatal functional deficits rather than developmental malformations.

Postnatal RDS with a lethal outcome can arise from several pathophysiological mechanisms that compromise neonatal lung function, including structural lung abnormalities, such as pulmonary hypoplasia or congenital alveolar dysplasia. Histological analyses of embryonic lungs at E15.5 and E17.5 revealed no abnormalities, confirming that lung morphogenesis proceeds normally in the absence of Gα_i2_. However, at birth, lungs of deceased neonates showed reduced alveolar surface area, thickened interalveolar septa, and increased mesenchymal tissue, consistent with impaired alveolar inflation and collapse. Despite macroscopically inflated lungs, electron microscopy demonstrated disrupted tubular myelin and inadequate surfactant organization at the air–liquid interface, while levels of saturated phosphatidylcholine and surfactant proteins were unaltered. This dissociation between normal surfactant production and impaired deployment highlights the complexity of surfactant biology in vivo.

The inability of surfactant to properly unfold and spread within alveoli, despite intact synthesis and secretion, suggests that Gα_i2_ plays a previously unknown regulatory role in the post-secretory phase of surfactant function rather than in the regulation of adequate production of pulmonary surfactant by alveolar type II (ATII) epithelial cells [30,34]. This agrees with the current understanding that several signaling pathways other than G_i_-dependent, including those involving G_s_ and G_q_ proteins, adenylyl cyclases, phospholipases, and protein kinases (PKA, PKC, and Ca^2+^/calmodulin-dependent kinases), have been implicated in surfactant secretion [41,42]. However, receptor-mediated and receptor-independent pathways involving G_i_ proteins have also been demonstrated to modulate surfactant release [43,44]. The transformation of lamellar body content into a functional monolayer and tubular myelin lattice is a multistep process dependent on biophysical and biochemical factors. Interestingly, the picture of large lamellar bodies and impaired surfactant processing is reminiscent of the pathophysiology of Hermansky–Pudlak syndrome (HPS) in humans. In this severe genetic disease, defective lysosome-related organelles lead to intracellular formation of giant lamellar bodies in AT II cells and accumulation of surfactant. In HPS pathophysiology, the chitinase-like-protein CHI3L1 and its impaired interplay with its receptor CRTH, a non-chemokine chemoattractant GPCR with a preference for Gα_i_ proteins, seems to play an important role, although the exact causal molecular pathways in HPS are still understudied [45]. Previous studies have shown that this process is influenced by the alveolar microenvironment, extracellular ionic composition, surfactant protein interactions, and alveolar lining fluid properties [30,35]. Our data suggest that Gα_i2_ signaling may modulate one or more of these factors, potentially by affecting epithelial–mesenchymal crosstalk, alveolar surface fluid composition, or calcium-regulated surfactant organization [41,42]. An alternative explanation for the respiratory failure in *Gnai2*-deficient neonates could be an impairment of neuromuscular control rather than an intrinsic pulmonary dysfunction. However, several observations argue against this possibility: *Gnai2*-deficient neonates were able to take initial deep breaths and exhibited normal spontaneous movements before death. Moreover, adult *Gnai2*-deficient mice showed no overt motor deficits, and lung function was normal or even slightly improved. Collectively, these findings make a neuromuscular origin unlikely and instead support a primarily lung-intrinsic mechanism, although secondary extrapulmonary effects cannot be fully excluded.

Interestingly, conditional deletion of *Gnai2* in alveolar type II (ATII) or mesenchymal cells did not replicate the lethal phenotype, suggesting involvement of other lung-resident cell types [46,47]. Endothelial cells and alveolar macrophages are known to influence surfactant turnover, alveolar surface tension, and inflammatory signaling—all of which could impair surfactant ultrastructure and spreading without altering its production. Dysregulated inflammatory mediators or calcium signaling pathways, both linked to G protein-coupled receptor activity, are plausible mechanisms underlying the observed defect [34]. The incomplete penetrance of the phenotype, with ~20% mortality among *Gnai2*-deficient neonates, suggests possible compensatory mechanisms and/or strain-specific genetic modifiers, as reported for other knockout models [48,49,50,51]. Differences in lethality between breeding strategies support this interpretation, though early procedural variability seems more likely. The lethality rate observed in the second study (~20%) aligns with the ~25% increase seen initially, including the unexpected mortality among heterozygous controls. Due to ethical constraints, these factors could not be fully disentangled.

We also observed a slight tendency toward increased kidney weight in deceased neonates (see Appendix A), possibly reflecting systemic responses to hypoxia, acute cardiopulmonary failure with fluid retention and congestion, or additional effects of *Gnai2*-deficiency in other organs. Although the significance of this finding remains speculative, it highlights the complex interaction between pulmonary and extrapulmonary factors during neonatal adaptation. Although, to the best of our knowledge, no germline mutations in *GNAI2* have been directly linked to lung disease in humans to date. Recent studies have identified *GNAI2* mutations, which lead to impaired GTPase activity of Gα_i2_ followed by an enhanced suppression of cAMP production and are associated with immune system dysregulation [52]. Despite these functions, *GNAI2* remains to be identified as a significant locus in recent genome-wide association studies of lung function traits [53].

Although approximately 20% of *Gnai2*-deficient mice died from postnatal respiratory failure due to immature lungs and insufficient surfactant unfolding, those that survived this critical period exhibited unexpectedly improved lung functions compared with wild-type controls, including higher flow rates, tidal volume, and compliance. Upregulation of the remaining Gα_i3_ isoform has previously been reported in various tissues of *Gnai2*-deficient mice [54,55], and likely represents a compensatory mechanism that may contribute to the enhanced pulmonary performance observed here. In addition to potential Gα_i3_-mediated compensation, other—yet unidentified—factors could also facilitate postnatal survival by functionally substituting for missing Gα_i2_ activity. While most G-protein signaling mechanisms are classically attributed to the α subunits, it is now well established that βγ dimers also mediate numerous cellular functions [1]. Thus, a possible contribution of βγ-dimer-dependent signaling to this complex compensatory network should also be considered.

Taken together, our results highlight a previously unappreciated regulatory role of Gα_i2_ in neonatal respiratory physiology. We propose that Gα_i2_ facilitates the structural organization of surfactant, likely through effects on the alveolar microenvironment and intercellular communication among epithelial, endothelial, mesenchymal, and immune cells. Further investigation into these pathways—including the roles of calcium signaling, inflammatory mediators, and genetic modifiers—will be essential to fully elucidate the mechanisms underlying this phenotype and may inform therapeutic approaches for neonatal RDS.

### 3.1. Conclusions

Gα_i2_ serves as a regulatory factor in neonatal respiratory adaptation by facilitating surfactant deployment and alveolar stabilization. Its deficiency leads to an incompletely penetrant phenotype, likely due to compensatory mechanisms and genetic modifiers. These findings emphasize the complexity of neonatal respiratory physiology and the need for precise regulation of surfactant function and intercellular communication.

### 3.2. Future Directions

Future research should elucidate how Gα_i2_ signaling integrates with epithelial, endothelial, mesenchymal, and immune pathways to regulate surfactant deployment and alveolar mechanics. Investigating calcium signaling, inflammatory mediators, and genetic modifiers will be crucial to understanding the incomplete phenotype. Such insights may pave the way for novel interventions in neonatal RDS and related disorders.

## 4. Materials and Methods

Experimental animals. The generation and phenotypic characterization of *Gnai2*-deficient mice (*Gnai2^−^^/−^*), has been described [13]. *Gnai2*-deficient mice were backcrossed on a C57BL6/N background for more than 11 generations. To study postnatal mortality of *Gnai2*-deficient mice, newborn homozygous *Gnai2*-deficient mice were intercrossed and compared to homozygous wild-type litters. For the generation and analysis of mesenchymal cell-specific *Gnai2*-deficient mice, floxed *Gnai2* (*Gnai2*^fl/fl^) mice were crossed with *Dermo*-Cre mice [38]. In addition, alveolar type II (ATII)-specific double-deficient floxed *Gnai2; Gnai3*-deficient mice (*Gnai2*^fl/fl^; *Gnai3*^+/fl^) [56] were crossed with *Shh*-Cre mice [39]. Absence of Gα_i2_ and Gα_i3_ protein was approved by immunoblot analysis (Appendix A).

For isolated lung perfusion experiments, 8-to-18-week old mice from homozygous and heterozygous breeding were analyzed. Animal experiments were conducted in accordance with the recommendations in the Guide for the Care and Use of Laboratory Animals (FELASA). Animals were kept in standard cages with enrichment under specific pathogen-free conditions (SPF) according to national guidelines for animal care at the animal facility of the University of Düsseldorf and Tübingen. All animals were kept with a 12 h light/dark cycle and had access to food and water ad libitum. The physical condition was monitored daily both before and during the experiments. No animal became severely ill or died before the experimental endpoint. Protocols were approved by the committee on the Ethics of Animal Experiments of local authorities. All surgeries were performed under anesthesia, and all efforts were made to minimize suffering. For IPL mice were anesthetized with Pentobarbital (120 mg/kg body weight).

Analysis of postnatal survival. Pregnancy in female mice was confirmed by the presence of a vaginal plug and designated as embryonic day 0.5 (E0.5). For cesarean section, pregnant females at gestational day 19.5 were sacrificed by cervical dislocation, whereas for spontaneous delivery, the pregnant mice were observed during the expected period of birth until the pups were born. In case of cesarian section, the abdominal cavity was opened by a transverse incision, and the uterine horns were excised. Fetuses were rapidly removed from the uterus and embryonic membranes. Spontaneous onset of labor typically occurred at E20.5. The postnatal survival of newborn mice was monitored for up to 120 min and correlated with their respiratory behavior. The time of death was defined as the point at which breathing ceased and respiratory arrest persisted without recovery.

Histology of lung tissue. For hematoxylin and eosin staining, lungs were removed from newborn mice, fixed in 4% formaldehyde, embedded in paraffin according to standard procedures, and longitudinal 6 µm serial sections were obtained. Alveolar and mesenchymal areas were quantified by hand. Morphometric analysis was performed using Adobe Photoshop (version Photoshop CC 2014) and ImageJ (version ImageJ2).

Electron Microscopy. For electron microscopy, lungs were fixed in 2.5% glutaraldehyde in 0.1 M cacodylate buffer (pH 7.4). The embedded lungs were sectioned into 1–2 mm^2^ small blocks and post-fixed in 1% OsO_4_ in 0.1 M cacodylate. For contrast enhancement, specimens were transferred to 70% ethanol saturated with 3% uranyl-acetate, dehydrated in a graded ethanol series, and embedded in Araldite. Ultrathin sections (50 nm) were contrasted with lead citrate and examined using a Zeiss electron microscope (EM 10CR).

Surfactant phospholipid analysis. Saturated phosphatidylcholine and total phospholipids were quantified from cell-free bronchoalveolar lavage fluid and whole lung tissue as previously described [57].

Isolated perfused mouse lung preparation. To determine adult mouse lung functions the Isolated Perfused Lung Size 1, Hugo-Sachs-Elektronik (HSE), March Hugstetten, Germany, was used as described [58]. Lungs were perfused in a non-circulating manner through the pulmonary artery at a constant flow of 1 mL/min. The perfusion medium (RPMI 1640) was supplemented with 1% L-Alanyl-L-Glutamine (200 mM), 4% Albumine bovine Fraction V and 0.12% NaCl. Lungs were ventilated by negative pressure (−8.5 to −3 cm H_2_O) with 90 breaths/min, resulting in a tidal volume in the range of 100 to 300 μL. A deep inflation (−25 cm H_2_O) was performed every 5 min to prevent the lungs from atelectasis. All data were transferred to a computer and analyzed by the Pulmodyn software (HSE) (version 1.1.1.202).

Immunoblot analyses. Dissected lungs were homogenized in lysis buffer containing 100 mM NaCl, 20 mM Tris–HCl, 2.5 mM EDTA and protease inhibitor (Complete Roche; Roche, Mannheim, Germany) to generate total protein lysates for subsequent immunoblot analyses. The proteins were separated by molecular weight via 12% SDS gels containing 6 M urea and subsequently transferred onto polyvinylidene difluoride membranes (PVDF; Merck Millipore, Darmstadt, Germany). The membranes were blocked in 5% milk–TBST (tris-buffered saline (TBS)–Tween 20), incubated with the indicated antibodies and developed with enhanced chemiluminescence (ECL; GE Healthcare, Buckinghamshire, UK). Gα_i2_ protein expression was visualized by immunodetection using either rabbit polyclonal anti-Gα_i1/i2_ or anti-Gα_i3_ antibodies (1:2000) and horseradish peroxidase (HRP)-conjugated anti-rabbit IgG secondary antibodies (1:2000) (Cell Signaling Technology, Danvers, MA, USA). Equal loading was verified with rabbit anti-α-tubulin (1:4000) (Cell signaling Technology, Cambridge, UK). Gα_i2_ and Gα_i3_ protein levels were quantified by densitometry using the ImageJ software (NIH, Bethesda, MD, USA) and were normalized to α-tubulin levels of the same samples.

Statistical analysis. Data are presented as means ± SED of individual data points. Data were analyzed using Students’ t-test for unpaired groups or one-way ANOVA following Bonferroni’s Multiple Comparison Test when comparing wild type, surviving and dead *Gnai2*-deficient mice. A *p*-value of less than 0.05 was considered statistically significant.

## Figures and Tables

**Figure 1 ijms-26-10655-f001:**
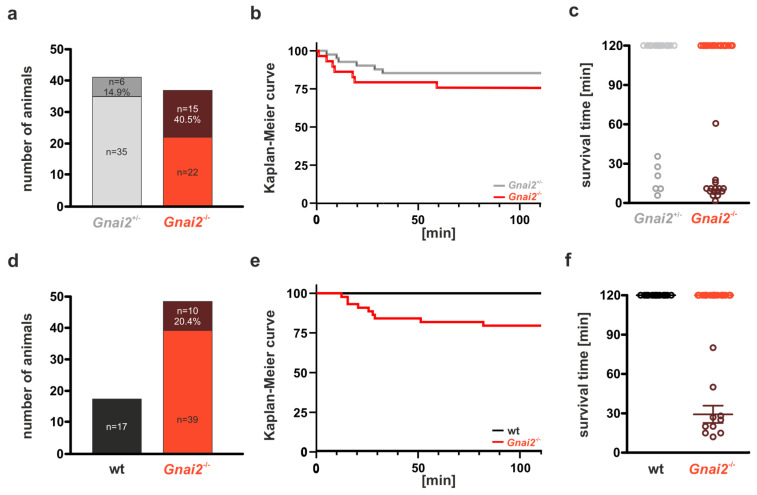
**Perinatal lethality of *Gnai2*-deficient newborns**. (**a**) Percentage of surviving (light grey, red) and deceased (grey, brown) *Gnai2*^+/−^ and *Gnai2*^−/−^ neonates delivered by cesarean section. (**b**) Kaplan–Meier survival curves of the corresponding *Gnai2*^+^/^−^ and *Gnai2*^−^/^−^ neonates over a 2 h observation period. (**c**) Individual survival times of *Gnai2*^+/−^ and *Gnai2*^−/−^ neonates. (**d**) Percentage of surviving (black, red) and deceased (brown) wild-type (wt) and *Gnai2*^-/-^ neonates after spontaneous delivery. (**e**) Kaplan–Meier survival curves of wt and *Gnai2*^−^/^−^ neonates showing a significant difference (log-rank test, *p* < 0.05) within the 2 h observation period. (**f**) Individual survival times of wt and *Gnai2*^−/−^ neonates. Exact time points of death for *Gnai2*^−^/^−^ neonates are indicated as brown data points in panels (**c**) and (**f**), while surviving *Gnai2*^−^/^−^ neonates are shown in red.

**Figure 2 ijms-26-10655-f002:**
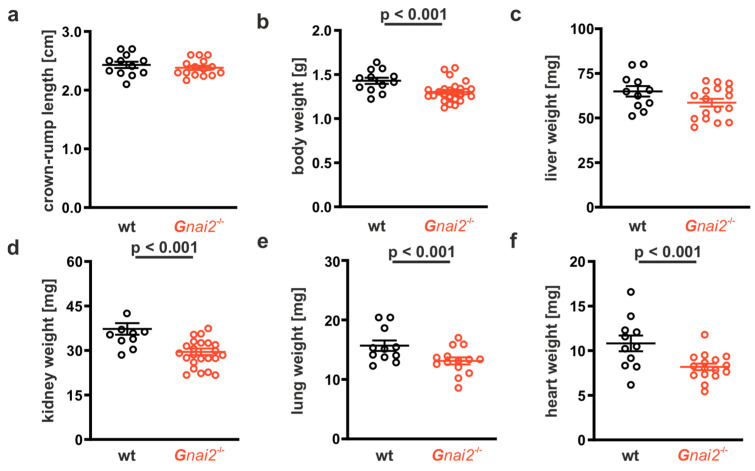
**Somatometric and organometric parameters of newborn mice**. (**a**) Crown–rump length was comparable between wild-type (wt) and *Gnai2*^−^/^−^ neonates. (**b**) Body weight was significantly reduced in *Gnai2*^−^/^−^ compared with wt neonates. (**c**) Liver weight was similar between genotypes. (**d**–**f**) Kidney, lung, and heart weights were significantly lower in *Gnai2*^−^/^−^ neonates than in wt controls. Data represent mean ± SEM from 12–21 animals per group.

**Figure 3 ijms-26-10655-f003:**
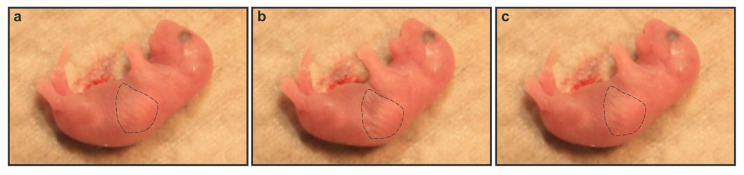
**Sequential phases of a respiratory cycle in a mouse**. Representative images illustrating distinct stages of a respiratory cycle during spontaneous breathing of a newborn *Gnai2*-deficient mouse that died: (**a**) before inspiration—the skin below the rib cage (black dotted line) appears smooth with minimal folding, (**b**) deep inspiration—prominent skin folds become visible beneath the rib cage, reflecting thoracic movement and negative intrathoracic pressure, (**c**) after expiration—the skin below the rib cage appears again smooth with minimal folding.

**Figure 4 ijms-26-10655-f004:**
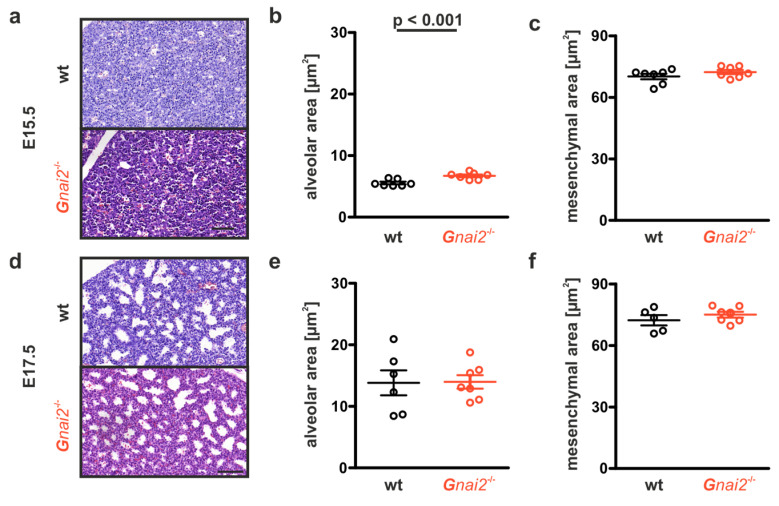
**Lung tissue structure of wt and *Gnai2*-deficient embryonic lungs at E15.5 and E17.5.** (**a**) Representative H&E staining of paraffin-embedded 5 µm lung sections obtained from whole body sections of wt and *Gnai2*-deficient embryos at E15.5. Quantification of (**b**) alveolar and (**c**) mesenchymal area derived from wt and *Gnai2*-deficient E15.5 embryonic lungs. (**d**) Representative H&E- stained lung sections from E17.5 wt and *Gnai2*-deficient embryos. Statistical analysis of (**e**) alveolar and (**f**) mesenchymal area quantified from E17.7 wt and *Gnai2*-deficient lungs. Data are mean ± SEM from 5 to 7 mice. Scale bars 100 µm.

**Figure 5 ijms-26-10655-f005:**
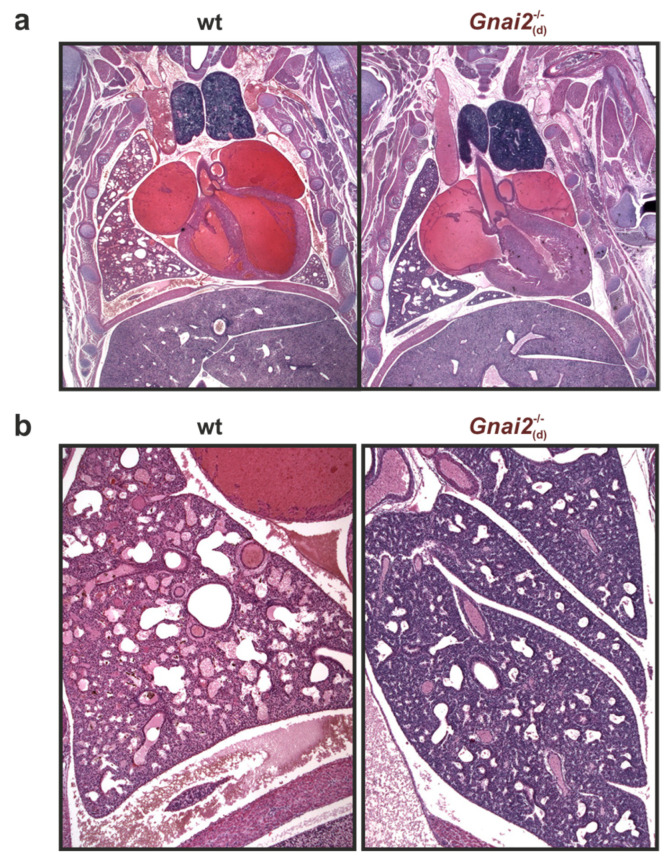
**Lung tissue structure of wt and deceased *Gnai2*-deficient (*Gnai**2*^−/−^_(d)_) neonates.** (**a**) Representative H&E staining of paraffin-embedded 5 µm thorax sections obtained from a wt newborn that survived the 2 h observation period and a deceased *Gnai2*-deficient neonate. (**b**) Representative H&E-stained lung sections from wt and deceased *Gnai2*-deficient neonates.

**Figure 6 ijms-26-10655-f006:**
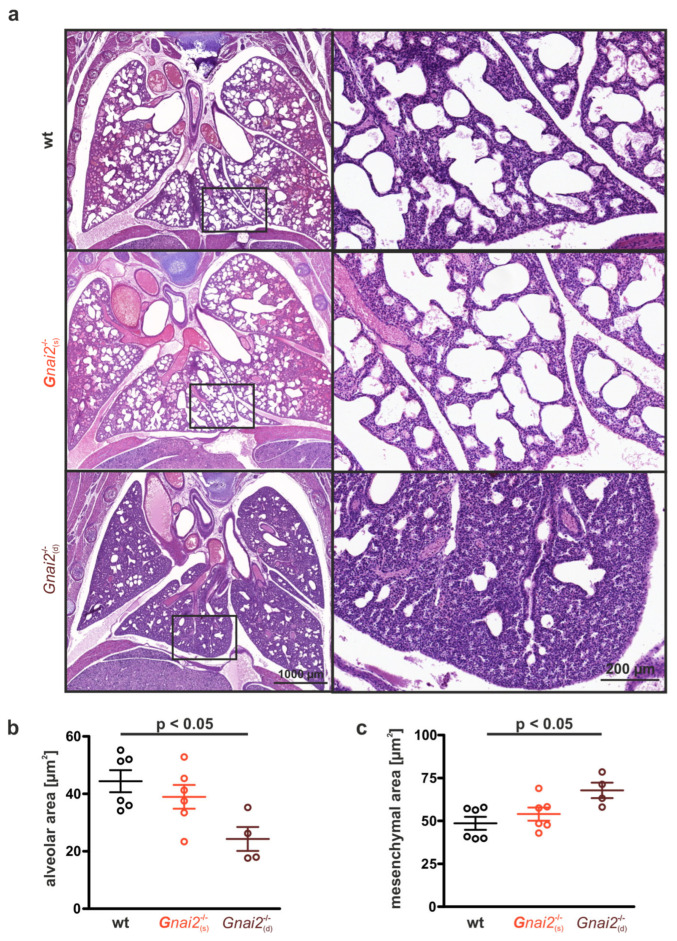
**Morphometric structure of lung tissue from *Gnai2*^−^^/^^−^_(d)_ mice at P0.** (**a**) Representative H&E staining of lung sections from wt, surviving *Gnai2*^−/−^_(s)_ and deceased *Gnai2*^−/−^_(d)_ animals. Scale bar 1000 µm (**left panel**). Magnifications (**right panel**) show significantly reduced alveolar areas in *Gnai2*^−/−^_(d)_ lungs. Scale bar 200 µm. Quantification of (**b**) alveolar and (**c**) mesenchymal area of wt (black), *Gnai2*^−/−^_(s)_ (red) and *Gnai2*^−/−^_(d)_ (brown) lungs. *Gnai2*^−/−^_(d)_ lungs have significantly decreased alveolar and significantly increased mesenchymal area. Mean ± SEM.

**Figure 7 ijms-26-10655-f007:**
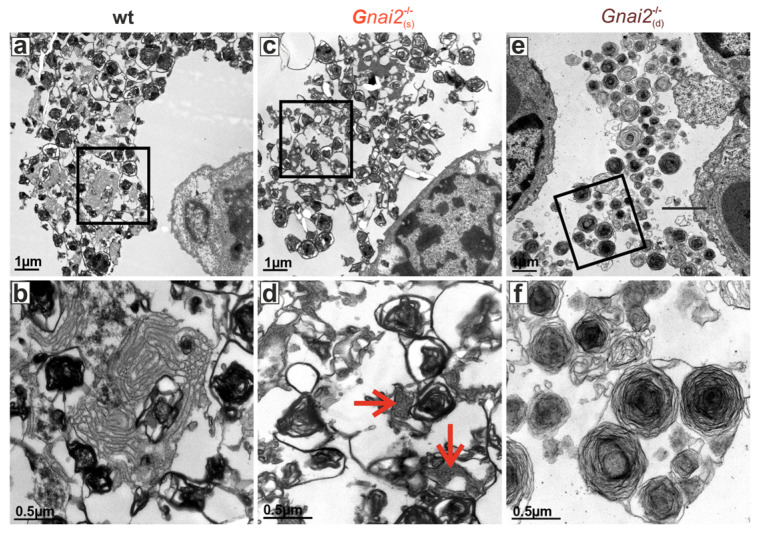
**Surfactant structure at P0**. Representative ultra-thin sections of lung tissue from wt (**left, a, b**), *Gnai2*^−/−^_(s)_ (**middle**, **c**,**d**) and *Gnai2*^−/−^_(d)_ (**right**, **e**,**f**) mice (**upper panel**). Scale bars 1 µm and 0.5 µm. Lower panels show magnifications of secreted surfactant and the newly formed tubular myelin (TM). Surfactant of *Gnai2*^−/−^_(d)_ lungs does not unravel as shown for wt and *Gnai2*^−/−^_(s)_ surfactant. In the upper panels (**a**,**c**,**e**), regions outlined by black boxes indicate areas that are shown at higher magnification in the lower panels (**b**,**d**,**f**) below.Red arrows in (**d**) indicate unraveled surfactant. Scale bars 1 and 0.5 µm.

**Figure 8 ijms-26-10655-f008:**
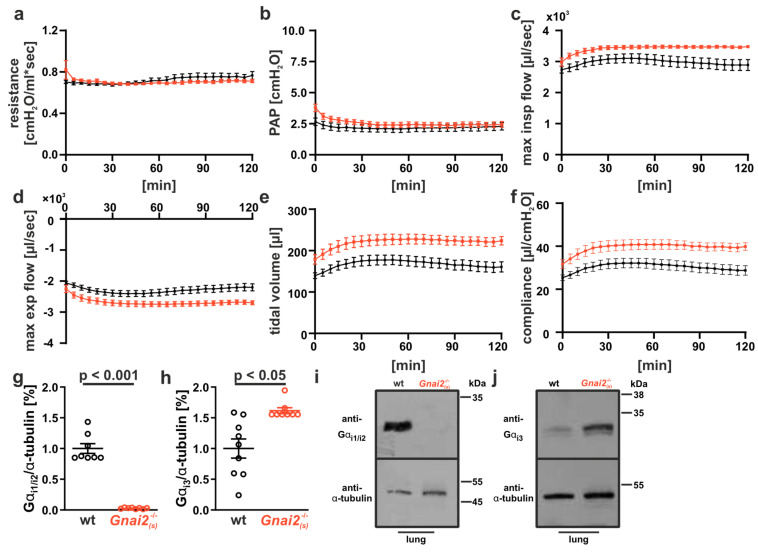
**Lung function of adult *Gnai2*^−^^/^^−^_(s)_ in comparison to wt animals analyzed by isolated perfused lung (IPL). ***Gnai2*-deficient mice (red) have normal (**a**) airway resistance and (**b**) pulmonary artery pressure (PAP), whereas (**c**) maximal inspiratory, (**d**) maximal expiratory flow rate, (**e**) tidal volume and (**f**) lung compliance were significantly improved compared to wt (black) animals. Significances for AUC levels are shown in Appendix A. Protein levels of (**g**) Gα_i2_ and (**h**) Gα_i3_ in lung tissue from wt and *Gnai2*-deficient mice. Protein levels of Gα_i3_ were significantly higher in *Gnai2*-deficient mice. Representative immunoblots for (**i**) Gα_i2_ and (**j**) Gα_i3_ expression in lung tissue from wt and *Gnai2*-deficient mice are depicted. Data are mean ± SEM from 9 to 12 animals.

**Table 1 ijms-26-10655-t001:** Mendelian ratio of littermates born from heterozygous *Gnai2*-matings. Statistics are based on Chi-square analyses (n.s.—not significant).

Genotype	E14	E18	P5-10	P21	% Expected
** *Gnai2* ^+/+^ **	7 (15.9%)	13 (22%)	137 (26.9%)	365 (31.1%)	25
** *Gnai2* ^+/−^ **	29 (66%)	34 (57.7%)	316 (62.1%)	695 (59.2%)	50
** *Gnai2* ^−/−^ **	8 (18.1%)	12 (20.3%)	56 (11%)	114 (9.7%)	25
	44 (6 litter)	59 (8 litter)	509 (92 litter)	1174 (210 litter)	
***p* value**	n.s.	n.s.	<0.001	<0.001	

## Data Availability

Research data will be shared on request.

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
