# Peer review of "Gαi2 Signaling Regulates Neonatal Respiratory Adaptation"

_ijms, 2025, doi:10.3390/ijms262110655_

Round 1
Reviewer 1 Report
Comments and Suggestions for Authors
In Leiss et al.’s manuscript “Gαi2 signalling regulates neonatal respiratory adaptation,” the authors provide an interesting new piece revealing post-natal lung-specific functions for Gαi2. Building from their previous work that noted a 50% mortality rate of Gαi2 -/- mice somewhere between E14 and P21, the authors sought to identify the exact timing and source of the lesion. After confirming the Mendelian ratio is lost in Gαi2 -/- mice and narrowing the window to between E18 and P5, they decided to test whether the mortality was either a result of issues in the birthing process or post-natal. Upon delivering the embryos via c-section and comparing WT vs Het vs KO, they observed that all the mice appeared healthy with no obvious physiological defects, supportive of a model of them having post-natal defects. They did observe, though, 50% mortality of Gαi2-/- mice within minutes after birth due to an inability to breathe, with apparent defects in the mouse respiratory cycle.
To determine the source of the respiratory defect, the researchers did a thorough analysis to try to identify the physiological origin of the defect. They observed a significant decrease in body weight as well as (in Fig. S2) a significant decrease in the kidney, lung, and heart weight (but not liver) of the Gαi2 -/- mice. They went on to do a thorough histological analysis of the tissue and observed different stages of embryonic lung development in WT vs KO. At E15.5, there appears to be a delay in the growth of the alveolar area but which later recovers by E17.5. While the postnatal survivors appear to have fairly similar lung morphology, the dead mouse lungs appear to have increased mesenchymal area and decreased alveolar area. Furthermore, they note a defect in surfactant structure in the KO mice that failed to survive. Yet, when they tried to create cell-type specific (mesenchymal versus ATII) knockout mice to see if they could specifically reproduce the lung defect, they could not reproduce the defect. These, and other analyses of phospholipid and SatPC content, suggest it is unlikely a primary defect in surfactant secretion.
The authors provide pivotal new insight into functions for Gαi2 in the context of lung function. Their work is thorough, and sets the stage for future work to dive into the mechanism by which Gαi2 regulates lung function. There are some ways that the authors could strengthen the work:
Major Comments:
- It seems like the writers should expand the possibilities as to what are the potential sources for the lung defect. In their discussion, they focused only on the lung itself and didn’t consider the possibility of other organ systems impinging and therefore being the source of the defect. A defect in neurological signaling or muscular function could explain the failure, and also explains the lung morphology. The fact that the live Gαi2-/- and the WT mouse tissues look fairly similar could simply be because that the mice succeeded in breathing while the dead mouse did not, so the lung could not expand with air. Therefore, this could potentially explain why the mesenchymal cells look dense, the alveoli look small, and the lung mass is the same weight between the live versus dead mice. One just succeeded with expanding up with air and making the tissue expand and have large aveolae, and the other remained the original, compressed morphology. This is also supported by the fact that the authors show that by around E18 the lung tissue look fairly similar.
If the authors could comment on other models besides a disruption in lung physiology as the underlying cause of the lung defect (e.g, the involvement of neurological or muscular system defects) inside the Discussion, this would be very useful for those who may want to follow up on this very important finding.
- There is a mention that the lack of complete penetrance could be because of compensation by other Galphas. It would be interesting to test by Western blot to see if the expression levels of either of the other Gαi’s (or other Gα family members) has a change in expression that could potentially support that model.
Minor Comments:
- Line 39 “impressing” should be “impressive”
- Figure 2 is missing a WT breathing control as a source of comparison. It also would be useful to include time points.
- Supplementary Figure S2 should be included in the main text, because it is a critical and significant finding consistent with the rest of the data. And the logic for including Figure 3C is confusing. If I am reading the graph right, total WT vs total KO was significant in difference for organs, but for differentiating between dead versus alive KO, you only mention kidney as being different between the two KO conditions. No significant difference with WT is noted in this graph. If there is, the authors should indicate the significance on this graph, too. If not, why are the data no longer significant when distinguishing between WT and KO?
- As a followup of this point, it might be better to include the weight data before moving into the lung data, making this Figure 2 instead of Figure 3. It works better with the flow and would set the stage for bringing up lungs in Figure 3.
- Line 190- for novices to the field, you might want to describe what pseudoglandular means and canicular, and describe what the lung normally looks like in these stages. A diagram indicating what would be expected for these developmental stages would be helpful.
- For the data shown in Figure 5, please indicate if the KO is dead or survivor.
- (Line 230)- which histological data suggest the surfactant defect? It doesn’t seem clear from the data shown before this point, as the data supporting this model aren’t shown until later in the document. If you are trying to make a transition sentence, maybe instead phrase it as, “To determine what is the source of the breathing defect, we decided to test one hypothesis that defect surfactant secretion is the driver.”
- Cite the papers previously showing that Cre-SHH and Cre-Dermo1 are tissue specific drivers. It is surprising in particular that SHH is tissue specific, as it is expressed in so many different tissues…
- For the homozygous knockouts that live to full term, do they show any lung defects at all? How is their blood oxygen levels etc?
- Comment whether any mutations in Gαi2 have been found in humans that are suggestive of a lung connection, if any have been found yet.
- Arrows in Fig 7 are black- maybe make them a different color so that they are easier to see on the black and white image? Also mention what the arrows are pointing to in the figure legend.
Author Response
In Leiss et al.’s manuscript “Gαi2 signalling regulates neonatal respiratory adaptation,” the authors provide an interesting new piece revealing post-natal lung-specific functions for Gαi2. Building from their previous work that noted a 50% mortality rate of Gαi2 -/- mice somewhere between E14 and P21, the authors sought to identify the exact timing and source of the lesion. After confirming the Mendelian ratio is lost in Gαi2 -/- mice and narrowing the window to between E18 and P5, they decided to test whether the mortality was either a result of issues in the birthing process or post-natal. Upon delivering the embryos via c-section and comparing WT vs Het vs KO, they observed that all the mice appeared healthy with no obvious physiological defects, supportive of a model of them having post-natal defects. They did observe, though, 50% mortality of Gαi2-/- mice within minutes after birth due to an inability to breathe, with apparent defects in the mouse respiratory cycle.
To determine the source of the respiratory defect, the researchers did a thorough analysis to try to identify the physiological origin of the defect. They observed a significant decrease in body weight as well as (in Fig. S2) a significant decrease in the kidney, lung, and heart weight (but not liver) of the Gαi2 -/- mice. They went on to do a thorough histological analysis of the tissue and observed different stages of embryonic lung development in WT vs KO. At E15.5, there appears to be a delay in the growth of the alveolar area but which later recovers by E17.5. While the postnatal survivors appear to have fairly similar lung morphology, the dead mouse lungs appear to have increased mesenchymal area and decreased alveolar area. Furthermore, they note a defect in surfactant structure in the KO mice that failed to survive. Yet, when they tried to create cell-type specific (mesenchymal versus ATII) knockout mice to see if they could specifically reproduce the lung defect, they could not reproduce the defect. These, and other analyses of phospholipid and SatPC content, suggest it is unlikely a primary defect in surfactant secretion.
The authors provide pivotal new insight into functions for Gαi2 in the context of lung function. Their work is thorough, and sets the stage for future work to dive into the mechanism by which Gαi2 regulates lung function. There are some ways that the authors could strengthen the work:
Major Comments:
- It seems like the writers should expand the possibilities as to what are the potential sources for the lung defect. In their discussion, they focused only on the lung itself and didn’t consider the possibility of other organ systems impinging and therefore being the source of the defect. A defect in neurological signaling or muscular function could explain the failure, and also explains the lung morphology. The fact that the live Gαi2-/- and the WT mouse tissues look fairly similar could simply be because that the mice succeeded in breathing while the dead mouse did not, so the lung could not expand with air. Therefore, this could potentially explain why the mesenchymal cells look dense, the alveoli look small, and the lung mass is the same weight between the live versus dead mice. One just succeeded with expanding up with air and making the tissue expand and have large aveolae, and the other remained the original, compressed morphology. This is also supported by the fact that the authors show that by around E18 the lung tissue look fairly similar.
If the authors could comment on other models besides a disruption in lung physiology as the underlying cause of the lung defect (e.g, the involvement of neurological or muscular system defects) inside the Discussion, this would be very useful for those who may want to follow up on this very important finding.
We appreciate the reviewer’s insightful comment and concur that alternative mechanisms underlying the observed respiratory failure merit consideration. Accordingly, we have expanded the Discussion to address potential neuromuscular and neuronal contributions (p. 11 of the revised manuscript). Notably, deceased Gnai2-deficient neonates were able to take an initial deep breath (see Fig. 3 of the revised manuscript) but failed to transition into the regular rapid breathing pattern, despite exhibiting normal spontaneous movement prior to death. Although a modest reduction in skeletal muscle mass could not be excluded previously (PMID: 24298018), our current data demonstrate normal or even improved lung function in surviving adults. In line with this, Gnai2-deficient mice at older age examined showed no increased mortality compared with wt controls (PMID: 23225882). Together, these findings make a neuromuscular or neuronal cause is unlikely and instead support a predominantly lung-intrinsic mechanism, while secondary extrapulmonary effects cannot be entirely ruled out.
- There is a mention that the lack of complete penetrance could be because of compensation by other Galphas. It would be interesting to test by Western blot to see if the expression levels of either of the other Gαi’s (or other Gα family members) has a change in expression that could potentially support that model.
We thank the reviewer for this insightful suggestion. Indeed, we observed an upregulation of Gαi3 in the lungs of adult mice, as shown in the newly added Figure 8. This is consistent with previous studies examining other tissues, which have reported similar compensatory upregulation of Gαi3. While this observation supports the assumption of partial compensation by other Gαi family members, further experiments would be required to establish a direct causal link in the context of lung function.
Minor Comments:
- Line 39 “impressing” should be “impressive” –
Done.
- Figure 2 is missing a WT breathing control as a source of comparison. It also would be useful to include time points.
Thank you for this valuable comment. We have addressed this point in detail in the revised manuscript by expanding the description of the physiological breathing pattern in wild-type (wt) neonates and the pathological abnormalities observed in Gnai2–/– newborns that died shortly after birth (e.g. p. 4, first paragraph, p. 5, last paragraph). The previous Figure 2 (now Figure 3) exemplifies the situation in affected Gnai2–/– mice. As shown in panels 3a–c, these animals initially displayed deep inspiratory efforts with subcostal and abdominal retractions, consistent with normal postnatal respiratory activity, indicating intact respiratory muscle function. Therefore, we considered parallel wt images unnecessary for direct comparison. In contrast to the physiological situation, in deceased Gnai2–/– neonates the gasping persisted until death.
Supplementary Figure S2 should be included in the main text, because it is a critical and significant finding consistent with the rest of the data. And the logic for including Figure 3C is confusing. If I am reading the graph right, total WT vs total KO was significant in difference for organs, but for differentiating between dead versus alive KO, you only mention kidney as being different between the two KO conditions. No significant difference with WT is noted in this graph. If there is, the authors should indicate the significance on this graph, too. If not, why are the data no longer significant when distinguishing between WT and KO?
We thank the reviewer for highlighting this point. In response, Supplementary Figure S2 has been now moved to the main manuscript (now new Figure 2) to reflect its importance and consistency with the rest of the data. To clarify, Figure 2 now shows the comparison of wt versus all Gnai2-/- mice pooled together, and a t-test confirms significant differences for all parameters, except the liver. Conversely, original Figure 3 has been moved to the Supplementary material (new Suppl. Figure 1), where three groups are compared: wt, surviving Gnai2-/-, and deceased Gnai2-/- mice. When the Gnai2-/- mice are subdivided into these two subgroups, an ANOVA is required for statistical comparison. In this analysis, no significant differences are observed. This is likely due to the smaller group sizes and increased variance among the three groups, which reduces statistical power, rather than a lack of biological effect. To avoid confusion and to prevent combining different statistical tests within a single figure, we have now removed the direct comparison of kidney between survived Gnai2-/- and deceased Gnai2-/- mice. Importantly, the overall conclusions regarding differences between wt and KO mice remain unchanged.
- As a followup of this point, it might be better to include the weight data before moving into the lung data, making this Figure 2 instead of Figure 3. It works better with the flow and would set the stage for bringing up lungs in Figure 3.
Thanks for this idea. Accordingly, Supplementary Figure 2 has been moved to the main manuscript as new Figure 2, whereas former Figure 3 is now part of the Supplemental material as new Supplemental Figure 1.
- Line 190- for novices to the field, you might want to describe what pseudoglandular means and canicular, and describe what the lung normally looks like in these stages. A diagram indicating what would be expected for these developmental stages would be helpful.
Thank you for pointing this out. We have characterized these stages more clearly in the revised manuscript (see p. 4, first paragraph, and p. 5, last paragraph, and p. 6, last paragraph of the revised ms.).
- For the data shown in Figure 5, please indicate if the KO is dead or survivor.
We thank the reviewer for this helpful comment and apologize for not having made this sufficiently clear in the original version. The H&E staining presented in Figure 5 depicts lung sections from a wild-type (wt) mouse that survived the observation period and from a Gnai2–/– mouse that died during this time (indicated as “Gnai2-/-(d)” in the figure). The wt animal was euthanized after 120 minutes to enable histological examination of the lung tissue. This information has now been explicitly stated in the revised Figure Legend to ensure clarity.
- (Line 230)- which histological data suggest the surfactant defect? It doesn’t seem clear from the data shown before this point, as the data supporting this model aren’t shown until later in the document. If you are trying to make a transition sentence, maybe instead phrase it as, “To determine what is the source of the breathing defect, we decided to test one hypothesis that defect surfactant secretion is the driver.”
We thank the reviewer for this valuable suggestion. The corresponding passage was revised accordingly (p. 8, last paragraph) to clarify the transition and introduce the hypothesis that defective surfactant secretion may underlie the breathing defect.
- Cite the papers previously showing that Cre-SHH and Cre-Dermo1 are tissue specific drivers. It is surprising in particular that SHH is tissue specific, as it is expressed in so many different tissues…
We appreciate this insightful comment. Shh-Cre and Dermo1-Cre lines are well established in lung development research as efficient drivers for recombination in pulmonary epithelial and mesenchymal cells, respectively (PMID: 15315763, 16452165, 24058167, 12553906, 21750028, 30153454). The corresponding references have been added in the revised manuscript (p. 9, last paragraph). Although SHH is broadly expressed, the Cre-expressing Gnai2; Gnai3-deficient mice did not exhibit increased lethality (see Supplementary Table 5).
- For the homozygous knockouts that live to full term, do they show any lung defects at all? How is their blood oxygen levels etc?
We thank the reviewer for this important comment. We had already addressed this issue and found, unexpectedly, that adult Gnai2–/– mice exhibited improved lung function compared to wt controls when assessed in the isolated perfused lung (IPL) system. Although blood oxygen levels were not measured directly, these IPL data provide clear functional evidence for preserved and even enhanced pulmonary performance. In the revised manuscript, we have added a new Results section (p. 10) including a new Figure 8 and a new Supplementary Table 6 to present these findings. In addition, we have also addressed these findings in the Discussion section (p. 12, third paragraph) of the revised manuscript.
- Comment whether any mutations in Gαi2 have been found in humans that are suggestive of a lung connection, if any have been found yet.
We thank the reviewer for this valuable point. To the very best of our knowledge, there is no published evidence to date of germline mutations in GNAI2 directly linked to monogenic lung diseases in humans. However, several lines of evidence suggest that Gαi2 may be functionally relevant in both pulmonary biology and disease contexts.
There is now robust evidence for disease-causing activating GNAI2 mutations in humans with immune dysregulation syndromes that impair T cell trafficking and signaling, although no direct lung phenotype has been reported [PMID: 39298586]. Although large-scale genome-wide association studies (GWAS) of lung function have not identified GNAI2 as a distinct locus so far [PMID: 30804560], its known signaling functions, expression in immune and smooth muscle cells, support its broader relevance to lung physiology and pathology. We have now included this point in the revised discussion (see p. 12, second paragraph).
- Arrows in Fig 7 are black- maybe make them a different color so that they are easier to see on the black and white image? Also mention what the arrows are pointing to in the figure legend.
Thank you for this suggestion. We have changed the arrows in Figure 7 to a contrasting colour for better visibility and updated the figure legend to clearly indicate what the arrows are pointing to.

Reviewer 2 Report
Comments and Suggestions for Authors
The article "Gαi2 signalling regulates neonatal respiratory adaptation" by Leiss et al. focuses on the role of afforementioned Gαi2 in the prenatal and postnatal development of functioning of the respiratory system in mice. This is a well designed and written study, I have only few comments and suggestions.
- While the study focuses on the moments of postpartum life of WT and Gαi2 deficient mice, I would still like to see, how the surviving Gαi2 deficient mice would develop further, and what, if any, effect would the surfactant deficiency have on them, if it is possible, since this is knowledge would be of great interest if translated onto the human early RDS, and, perhaps, would inform a favourable course of therapy for such patients.
- A somewhat hypothetical question - would emergency introduction of surfactant to the lungs of failing Gαi2 deficient neonate mice alleviate the symptoms? I think this could be disscussed in the context of whether the Gαi2 deficiency affects mice through impaired surfactant synthesis, or there are some additional factors in the works.
Author Response
The article "Gαi2 signalling regulates neonatal respiratory adaptation" by Leiss et al. focuses on the role of afforementioned Gαi2 in the prenatal and postnatal development of functioning of the respiratory system in mice. This is a well designed and written study, I have only few comments and suggestions.
- While the study focuses on the moments of postpartum life of WT and Gαi2 deficient mice, I would still like to see, how the surviving Gαi2 deficient mice would develop further, and what, if any, effect would the surfactant deficiency have on them, if it is possible, since this is knowledge would be of great interest if translated onto the human early RDS, and, perhaps, would inform a favourable course of therapy for such patients.
Thank you for this important comment. In the revised manuscript we have now included the examination of the surviving Gnai2-deficient mice in adulthood (new section (p. 10) including new Figure 8 and new Supplementary Table 6). In addition, we have also addressed these findings in the Discussion section (p. 12) of the revised manuscript. The lung function of the surviving Gnai2-deficient mice was assessed using isolated perfused lungs (IPL), and we found that lung function was actually improved compared to the controls. Based on these IPL data, we conclude that the surviving Gnai2-deficient mice may not exhibit a persistent surfactant deficiency. These IPL data have now been included as new Figure 8 and Supplementary Table 6 in the revised manuscript. While a direct translation to human early respiratory distress syndrome (RDS) requires caution, these results suggest that the critical window for surfactant-related respiratory failure occurs immediately postpartum, highlighting the potential importance of early intervention. Please, see also our response to Reviewer 1.
- A somewhat hypothetical question - would emergency introduction of surfactant to the lungs of failing Gαi2 deficient neonate mice alleviate the symptoms? I think this could be disscussed in the context of whether the Gαi2 deficiency affects mice through impaired surfactant synthesis, or there are some additional factors in the works.
We thank the reviewer for this interesting and thoughtful hypothetical question. We did not perform experiments involving emergency surfactant administration in neonatal Gnai2-deficient mice and therefore cannot directly assess its potential therapeutic effect. Based on our histological and functional data, respiratory failure in deceased neonates is likely driven primarily by impaired surfactant unravelling. However, as emphasised in the manuscript, we cannot exclude the contribution of additional factors, including extrapulmonary effects. Moreover, our findings in surviving adult Gnai2-deficient mice, particularly the IPL analyses showing normal or even improved lung function, suggest that the critical window for surfactant-related respiratory failure is confined to the immediate postpartum.

Reviewer 3 Report
Comments and Suggestions for Authors
The manuscript "Gαi2 signalling regulates neonatal respiratory adaptation" presents compelling evidence that Gnai2 plays a critical role in neonatal respiratory adaptation, which is a previously unappreciated function of this G protein. The study addresses the partial lethality of Gnai2-deficient mice, as reported in the literature. Using a combination of phenotypic analysis, histology, TEM-based ultrastructural examination, and genetic rescue, the authors attribute the cause to impaired lung function immediately after birth. Their observation that a significant proportion of Gnai2-deficient neonates die shortly after birth with signs of respiratory distress, such as cyanosis and abnormal breathing, is supported by histological data revealing a reduced alveolar surface area and thickened septa. This provides a clear anatomical basis for respiratory failure and excludes the possibility that the authors' observations are the result of survivor bias. Furthermore, the authors report impairment of surfactant ultrastructure and organization in non-surviving pups, suggesting a role for Gnai2 in the functional deployment of surfactant rather than in its synthesis or secretion. The authors also ruled out the possibility that this was caused by a gross developmental effect of the Gnai2 knockout by evaluating embryonic-stage lungs. Additionally, they performed detailed somatometric and organometric analyses to address the possibility of other causes of the observed lethality. Overall, the conclusion that Gnai2 modulates the surfactant system is reasonable, given that some deficient mice survive with compensatory or redundant mechanisms.
Overall, the manuscript is well conceived and executed, and the conclusions are generally supported by the presented data. This work will be of significant interest to researchers and will make a substantial contribution to the field. However, certain details require clarification, which the authors must address:
First, the comparisons to the Mendelian ratio presented in Table 1 and Supplementary Tables 1 and 2 are useful, but it would be best to present a detailed survival curve with a corresponding analysis. Additionally, it would be helpful to reinforce the significance of the observed deviations from the expected and control pup distributions with a statistical analysis (Chi-square test, for example).
The act of breathing and attempting to inflate the lungs is key to interpreting the findings and attributing them to a high likelihood of abnormality in surfactant deployment. However, this part of the study lacks a convincing quantitative representation, and the fact that all dead Gnai2⁻/⁻ mice exhibited this pattern is only mentioned in passing. It would be useful for the authors to present the number of neonates with breathing abnormalities on a graph next to the graph showing the lethality rate to reinforce the notion that all of them attempted to breathe. Additionally, the importance of this finding could be more clearly expressed in the manuscript text.
The data in Figures 1b and 1c are presented incorrectly. The survival data are non-parametric and cannot be evaluated with the corresponding tools. Instead, the authors should present both graphs as two-hour survival Kaplan-Meier curves and perform a log-rank test. Additionally, the figure is poorly annotated. It is difficult to understand that the data in panel B resulted from a C-section.
The authors claim that their findings are independent of genetic background and apparently used C57BL/6 and Sv129 animals for this purpose; however, there is no supporting data.
The study is reinforced by the genetic rescue of the observed phenotype, but this part is presented extremely sketchily. The authors should provide clear proof of the efficacy and lung-specificity of the recombination and describe the number of animals analyzed. They should also explain why the Gnai3 floxed allele was employed in this study. There is no materials and methods section for this part.
Author Response
The manuscript "Gαi2 signalling regulates neonatal respiratory adaptation" presents compelling evidence that Gnai2 plays a critical role in neonatal respiratory adaptation, which is a previously unappreciated function of this G protein. The study addresses the partial lethality of Gnai2-deficient mice, as reported in the literature. Using a combination of phenotypic analysis, histology, TEM-based ultrastructural examination, and genetic rescue, the authors attribute the cause to impaired lung function immediately after birth. Their observation that a significant proportion of Gnai2-deficient neonates die shortly after birth with signs of respiratory distress, such as cyanosis and abnormal breathing, is supported by histological data revealing a reduced alveolar surface area and thickened septa. This provides a clear anatomical basis for respiratory failure and excludes the possibility that the authors' observations are the result of survivor bias. Furthermore, the authors report impairment of surfactant ultrastructure and organization in non-surviving pups, suggesting a role for Gnai2 in the functional deployment of surfactant rather than in its synthesis or secretion. The authors also ruled out the possibility that this was caused by a gross developmental effect of the Gnai2 knockout by evaluating embryonic-stage lungs. Additionally, they performed detailed somatometric and organometric analyses to address the possibility of other causes of the observed lethality. Overall, the conclusion that Gnai2 modulates the surfactant system is reasonable, given that some deficient mice survive with compensatory or redundant mechanisms.
Overall, the manuscript is well conceived and executed, and the conclusions are generally supported by the presented data. This work will be of significant interest to researchers and will make a substantial contribution to the field. However, certain details require clarification, which the authors must address:
First, the comparisons to the Mendelian ratio presented in Table 1 and Supplementary Tables 1 and 2 are useful, but it would be best to present a detailed survival curve with a corresponding analysis. Additionally, it would be helpful to reinforce the significance of the observed deviations from the expected and control pup distributions with a statistical analysis (Chi-square test, for example).
We thank the reviewer for this helpful suggestion. Chi-square analyses have now been performed for all three Mendelian ratios (see Table 1 and Supplementary Tables 2 and 3). In addition, detailed survival curves have been included in the revised version (see responses below).
The act of breathing and attempting to inflate the lungs is key to interpreting the findings and attributing them to a high likelihood of abnormality in surfactant deployment. However, this part of the study lacks a convincing quantitative representation, and the fact that all dead Gnai2⁻/⁻ mice exhibited this pattern is only mentioned in passing. It would be useful for the authors to present the number of neonates with breathing abnormalities on a graph next to the graph showing the lethality rate to reinforce the notion that all of them attempted to breathe. Additionally, the importance of this finding could be more clearly expressed in the manuscript text.
We thank the reviewer for this valuable and constructive suggestion. To improve clarity, we have revised Figures 1c and 1f marking in a distinct color the exact time points at which Gnai2-/- neonates died, thereby better visualizing the rapid course of lethality. In addition, the Results section was revised to explicitly state that the lungs of deceased Gnai2-/- neonates were initially fully inflated despite their markedly reduced survival time.
The data in Figures 1b and 1c are presented incorrectly. The survival data are non-parametric and cannot be evaluated with the corresponding tools. Instead, the authors should present both graphs as two-hour survival Kaplan-Meier curves and perform a log-rank test. Additionally, the figure is poorly annotated. It is difficult to understand that the data in panel B resulted from a C-section.
We thank the reviewer for this valuable comment and fully agree with the concern regarding the appropriate presentation of survival data. Following the suggestion, we have now included Kaplan–Meier survival curves with log-rank test as new panels (now shown as Figure 1b and 1e). The original Figures 1b and 1c were revised accordingly and are now presented as Figures 1c and 1f in the revised manuscript. To improve clarity, the legend of Figure 1 has been expanded, and the methodological differences between the two experimental approaches – caesarean section (Figures 1a–c) versus spontaneous delivery (Figures 1d–f) – are now clearly described in the figure legend and the Materials and Methods (p. 13).
The authors claim that their findings are independent of genetic background and apparently used C57BL/6 and Sv129 animals for this purpose; however, there is no supporting data.
We thank the reviewer for pointing this out. The relevant data for Gnai2-/- mice on a 129/Sv background at P21 have now been incorporated into Table 1 and Supplemental Table 2. These data reflect Mendelian ratios and indicate comparable survival to the C57BL/6 background at this time point. For clarity, we note that all other experimental data presented in the manuscript were generated exclusively in animals on a C57BL/6 background.
The study is reinforced by the genetic rescue of the observed phenotype, but this part is presented extremely sketchily. The authors should provide clear proof of the efficacy and lung-specificity of the recombination and describe the number of animals analyzed. They should also explain why the Gnai3 floxed allele was employed in this study. There is no materials and methods section for this part.
We thank the reviewer for this comment and appreciate the opportunity to clarify the rationale of these experiments. These studies were performed to further investigate the molecular mechanisms underlying postnatal lethality in Gnai2⁻/⁻ newborns. To this end, we generated conditional knockout lines lacking Gαi2 either in alveolar type II (ATII) cells (Shh-Cre; Gnai2fl/fl) or in lung mesenchymal cells (Dermo1-Cre; Gnai2fl/fl). Surprisingly, neither conditional knockout line displayed increased peri- or postnatal mortality. Considering the observed upregulation of Gαi3 protein in lungs of surviving adult Gnai2⁻/⁻ mice (new Figure 8), we additionally generated ATII cell–specific double knockouts (Shh-Cre; Gnai2fl/fl; Gnai3fl/fl). Our data suggest that loss of Gαi2 and Gαi3 in these cell populations alone is not sufficient to cause the neonatal lethality observed in the global knockout. The corresponding experimental procedures have now been added to the Materials and Methods section.

Round 2
Reviewer 3 Report
Comments and Suggestions for Authors
The authors have done an excellent job of addressing all the points raised in the previous review. I appreciate the detailed response and the conscientious manner in which the revisions were incorporated. The manuscript is now much stronger and can certainly be recommended for publication.
However, the authors should correct the mistake in Fig. 8h and i, where Gnai3 is incorrectly labelled as Gnai2.
Author Response
Prof. Dr. Ines Liebscher
Guest Editor
Journal: IJMS
Section: Biochemistry
Special Issue: G Protein-Coupled Receptors in Cell Signaling
2nd Revision ijms-3831308
Dear Professor Liebscher,
Please find enclosed our revised manuscript entitled “Gαi2 signalling regulates neonatal respiratory adaptation” for publication in the special issue of IJMS - Biochemistry.
Thank you for the kind opportunity to revise our manuscript once again.
Enclosed you will find the revised version as well as a detailed response to the comments of the 3rd Reviewer.
We hope that the manuscript is now acceptable for publication and look forward to hearing from you.
With best regards
Bernd Nürnberg
Response to Reviewer 3:
|
Yes |
Can be improved |
Must be improved |
Not applicable |
|
|
Does the introduction provide sufficient background and include all relevant references? |
(x) |
( ) |
( ) |
( ) |
|
Is the research design appropriate? |
(x) |
( ) |
( ) |
( ) |
|
Are the methods adequately described? |
(x) |
( ) |
( ) |
( ) |
|
Are the results clearly presented? |
(x) |
( ) |
( ) |
( ) |
|
Are the conclusions supported by the results? |
(x) |
( ) |
( ) |
( ) |
|
Are all figures and tables clear and well-presented? |
( ) |
( ) |
(x) |
( ) |
Comments and Suggestions for Authors
The authors have done an excellent job of addressing all the points raised in the previous review. I appreciate the detailed response and the conscientious manner in which the revisions were incorporated. The manuscript is now much stronger and can certainly be recommended for publication.
However, the authors should correct the mistake in Fig. 8h and i, where Gnai3 is incorrectly labelled as Gnai2.
Submission Date 05 August 2025
Date of this review 14 Oct 2025 21:55:47
Response:
We sincerely thank the reviewer for the positive assessment of our revised manuscript and for pointing out an apparent labeling error in Figure 8. We have revised the figure panels and its legend accordingly.
In the revised version, the immunoblot analyses (originally shown in Fig. 8g–i) now clearly distinguish between the loss of Gαi2 and the compensatory increase of Gαi3 protein expression in the lungs of Gnai2-deficient mice. To improve clarity, we have separated these results into two distinct panels: the absence of Gαi2 protein is now presented in the new Fig. 8i, while the increased expression of Gαi3 protein is shown in the new Fig. 8j.
Please note that the analyses in Fig. 8g and i were performed using an antibody directed against Gαi2, whereas the data for Gαi3 expression were obtained with a specific anti-Gαi3 antibody. The corresponding figure legend has been updated and expanded to reflect these details.
